# Applications of genetic-epigenetic tissue mapping for plasma DNA in prenatal testing, transplantation and oncology

Wanxia Gai[1,2,3], Ze Zhou[1,2], Sean Agbor-Enoh[4,5,6], Xiaodan Fan[7], Sheng Lian[7], Peiyong Jiang[1,2,3], Suk Hang Cheng[1,2], John Wong[8], Stephen L Chan[9], Moon Kyoo Jang[4,6], Yanqin Yang[4,6], Raymond HS Liang[10], Wai Kong Chan[11], Edmond SK Ma[11], Tak Y Leung[12], Rossa WK Chiu[1,2], Hannah Valantine[4,6], KC Allen Chan[1,2,3], YM Dennis Lo[1,2,3]*

[1]Li Ka Shing Institute of Health Sciences, The Chinese University of Hong Kong, Hong Kong, China; [2]Department of Chemical Pathology, The Chinese University of Hong Kong, Prince of Wales Hospital, Hong Kong, China; [3]State Key Laboratory of Translational Oncology, The Chinese University of Hong Kong, Hong Kong, China; [4]Genomic Research Alliance for Transplantation (GRAfT), Bethesda, United States; [5]Division of Pulmonary and Critical Care Medicine, The Johns Hopkins School of Medicine, Baltimore, United States; [6]Division of Intramural Research, National Heart, Lung and Blood Institute, Bethesda, United States; [7]Department of Statistics, The Chinese University of Hong Kong, Hong Kong, China; [8]Department of Surgery, The Chinese University of Hong Kong, Prince of Wales Hospital, Hong Kong, China; [9]Department of Clinical Oncology, The Chinese University of Hong Kong, Prince of Wales Hospital, Hong Kong, China; [10]Comprehensive Oncology Centre, Hong Kong Sanatorium & Hospital, Hong Kong, China; [11]Department of Pathology, Hong Kong Sanatorium & Hospital, Hong Kong, China; [12]Department of Obstetrics and Gynaecology, The Chinese University of Hong Kong, Prince of Wales Hospital, Hong Kong, China

*For correspondence: loym@cuhk.edu.hk

Reviewing editor: Tony Yuen,

**Abstract** We developed genetic-epigenetic tissue mapping (GETMap) to determine the tissue composition of plasma DNA carrying genetic variants not present in the constitutional genome through comparing their methylation profiles with relevant tissues. We validated this approach by showing that, in pregnant women, circulating DNA carrying fetal-specific alleles was entirely placenta-derived. In lung transplant recipients, we showed that, at 72 hr after transplantation, the lung contributed only a median of 17% to the plasma DNA carrying donor-specific alleles, and hematopoietic cells contributed a median of 78%. In hepatocellular cancer patients, the liver was identified as the predominant source of plasma DNA carrying tumor-specific mutations. In a pregnant woman with lymphoma, plasma DNA molecules carrying cancer mutations and fetal-specific alleles were accurately shown to be derived from the lymphocytes and placenta, respectively. Analysis of tissue origin for plasma DNA carrying genetic variants is potentially useful for noninvasive prenatal testing, transplantation monitoring, and cancer screening.

## Introduction

The circulation receives DNA from different tissues and organs within the body. The analysis of plasma DNA from specific tissues or organs is useful for revealing and monitoring the pathological processes in different tissues. In scenarios where the genetic composition of a target tissue or organ

is different from the host constitutional genome, plasma DNA carrying the tissue- or organ-specific variants can be used to identify DNA molecules released by the tissue or organ. For example, in pregnant women, plasma DNA carrying fetal-specific alleles can be used for prenatal analysis of the fetal genetic constitution (*Kitzman et al., 2012*; *Lo et al., 2010*). In organ transplant recipients, the concentrations of donor-specific DNA has been used to reflect the tissue damage associated with acute rejection (*De Vlaminck et al., 2014*; *De Vlaminck et al., 2015*; *Knight et al., 2019*; *Lo et al., 1998*; *Schütz et al., 2017*). Notably, immediately after organ transplantation, the plasma concentration of donor-derived DNA surges (*De Vlaminck et al., 2015*). Because of this initial surge, the analysis for donor-derived DNA has limited value in identifying graft rejection and infection during the first 60 days of transplantation (*De Vlaminck et al., 2015*). The exact mechanism of this initial surge is unclear. It is possible that the hematopoietic cells within the transplanted organ are more likely to release a significant amount of DNA into the circulation during the initial days after the transplantation. However, existing methods for detecting DNA derived from a transplanted organ in plasma rely on identifying genetic differences between the organ donor and the recipient (*De Vlaminck et al., 2014*; *De Vlaminck et al., 2015*; *Knight et al., 2019*; *Lo et al., 1998*; *Schütz et al., 2017*). These methods cannot be used to further distinguish the exact cell types the donor DNA is derived from.

In situations where the genetic compositions of the different organs are the same, tissue composition analysis based on detecting organ-specific alleles would not be applicable. To overcome this, recent efforts have been made to measure the composition of DNA using epigenetic approaches. These approaches include methylation deconvolution (*Moss et al., 2018*; *Sun et al., 2015*), mapping nucleosomal patterns (*Snyder et al., 2016*; *Sun et al., 2019*), analysis of end DNA motifs, end positions and jaggedness (*Chan et al., 2016*; *Jiang et al., 2018*; *Jiang et al., 2020b*; *Jiang et al., 2020a*), and the profiling of RNA transcripts (*Koh et al., 2014*; *Tsui et al., 2014*). In these methods, the features of interest, for example, methylation patterns, of the plasma DNA were profiled and compared with those of the candidate tissues. Then the relative contribution of the different tissues to the circulating DNA was determined mathematically. One potential application of plasma DNA tissue composition analysis is to reveal the likely location of a concealed cancer. Recently, it has been shown that the analysis for circulating cell-free tumor DNA (ctDNA) is useful for the screening of early asymptomatic cancers (*Chan et al., 2017*; *Lennon et al., 2020*; *CCGA Consortium et al., 2020*). As cancer-associated genetic and epigenetic changes are present in virtually all types of cancers (*Chan et al., 2013a*; *Chan et al., 2013b*; *Leary et al., 2012*; *Wong et al., 1999*), the detection of these cancer-associated aberrations in plasma can potentially serve as universal tumor markers for the screening of cancers in general. However, how subjects with positive results of a universal cancer test can be further worked up is an important but relatively under-explored topic. In a study by Lennon et al., subjects tested positive with ctDNA test that detected a wide variety of cancers were investigated with whole body positron emission tomography-computed tomography (PET-CT) (*Lennon et al., 2020*). If the potential tissue origin of the cancer can be obtained from ctDNA analysis, more focused investigations, for example, high-resolution imaging of an affected organ, can be performed. These organ-specific investigations could provide better sensitivity and specificity and could be achieved with a lower dose of radiation to the patients. In previous proof-of-principle studies, the tissue origin of cancers was successfully revealed by plasma DNA deconvolution (*CCGA Consortium et al., 2020*; *Moss et al., 2018*; *Sun et al., 2015*). However, existing approaches only allow tissue composition analysis of the whole pool circulating DNA rather than specifically to the tumor-derived DNA. The accuracy of these approaches would be affected by the fractional concentration of tumor-derived DNA in the sample.

In this study, we developed a method called genetic-epigenetic tissue mapping (GETMap) to determine the tissue composition of plasma DNA carrying genetic variants which are different from the host constitutional genome. This method is based on the comparison of the methylation profiles of the plasma DNA carrying genetic variants and the relevant tissues or organs that plasma DNA is potentially derived from. First, we validated this approach using a pregnancy model through the analysis of the tissue origin of the plasma DNA carrying fetal-specific alleles. Then, we applied this method to measure the tissue compositions of plasma DNA carrying cancer-associated mutations (i. e., present in tumor cells or plasma but absent from buffy coats) in hepatocellular cancer (HCC) patients and those molecules carrying donor-specific alleles in lung transplant recipients. The former analysis can provide information regarding the tissue origin of the cancer and the latter analysis

provided insights on the reason for the surge of donor-derived DNA in the plasma of organ transplant recipients during the early post-transplantation period.

## Results

### Principle of GETMap

The principle of the GETMap analysis is illustrated in *Figure 1*. The first step is to identify different sets of plasma DNA molecules based on genotypic differences. For example, the two sets of plasma DNA molecules carrying cancer-associated mutations and wildtype alleles were identified in cancer patients. In organ transplant recipients, three sets of DNA molecules can be identified, including those carrying the host-specific, recipient-specific alleles and alleles shared between the host and recipient. Similarly, three sets of molecules could be identified in the plasma of a pregnant woman, namely those carrying fetal-specific, maternal-specific alleles and alleles shared by the mother and fetus. Then, the tissue compositions were determined for each set of plasma DNA molecules through comparing the methylation profile of the plasma DNA molecules and the methylation profiles of the relevant tissues after bisulfite sequencing. While there are some similarities between the deconvolution step and that described in our previous study (*Sun et al., 2015*), there are notable differences. First, only DNA molecules of interest, for example, those carrying fetal-specific alleles, or cancer-associated mutations or donor-derived alleles, are analyzed. Second, only CpG sites near informative single nucleotide polymorphism (SNP) alleles are included in the algorithm. The details of the mathematical calculation are described in the 'Materials and methods' section. For the choice of candidate tissues used for the GETMap analysis, we included the tissues (including neutrophils, lymphocytes, liver, and placenta) that have been validated in a previous study on tissue deconvolution by methylation analysis (*Sun et al., 2015*). The inclusion of the placenta also allows us to use the analysis of fetal DNA in maternal plasma as a model to validate this new approach. As this study also analyzed patients receiving lung transplantation, lung is further included as one candidate tissue in the plasma DNA deconvolution. The methylation status of the plasma DNA molecules was determined by bisulfite sequencing.

### Accuracy of GETMap analysis

To evaluate the accuracy of our approach, we performed simulation analyses using GETMap to deconvolute five types of reference tissues including neutrophils, lymphocytes, lung, liver, and placenta. Three sets of simulation analyses were performed to simulate the three clinical application scenarios in our study, namely pregnancy, transplantation, and cancer detection. For each scenario, the numbers of informative DNA fragments, CpG sites, and sequencing depth were matched with the median of the studied samples. Thirty independent simulations were performed for each scenario. The accuracy was calculated as the percentage contribution assigned to the tissue used for the deconvolution. For example, when the bisulfite sequencing data of liver tissue is used for deconvolution, the accuracy would refer to the estimated contribution from liver. The median accuracy of GETMap analyses for reference tissues was 98.3% (range 95.5–99.8%) (*Table 1*).

### Deconvolution of fetal- and maternal-derived DNA in maternal plasma

We first used the analysis of plasma DNA of pregnant women as a model to demonstrate the feasibility of GETMap. Venous blood samples were collected from 30 pregnant women with 10 in each of the first, second, or third trimesters of gestation. Placental tissues were obtained from chorionic villus sampling or amniocentesis for the first and second trimester pregnant women. For third trimester pregnant women, the placenta was collected after delivery. The pregnant woman and the placental tissue were genotyped using the Illumina whole-genome arrays (HumanOmni2.5, Illumina). Based on the genotypes of the mother and fetus, we identified a median of 189,862 (range 14,035–192,998) maternal-specific informative SNPs where the mother was heterozygous and the fetus was homozygous, and a median of 194,479 (range 145,743–201,847) fetal-specific informative SNPs where the mother was homozygous and the fetus was heterozygous. After bisulfite sequencing of maternal plasma DNA, a median of 103 million uniquely mapped reads (range: 52–186 million) were identified in the maternal plasma DNA samples. Plasma DNA molecules carrying the fetal- and maternal-specific alleles were identified. A median of 162,813 CpG sites (range 8237–295,671) and 53,039 CpG

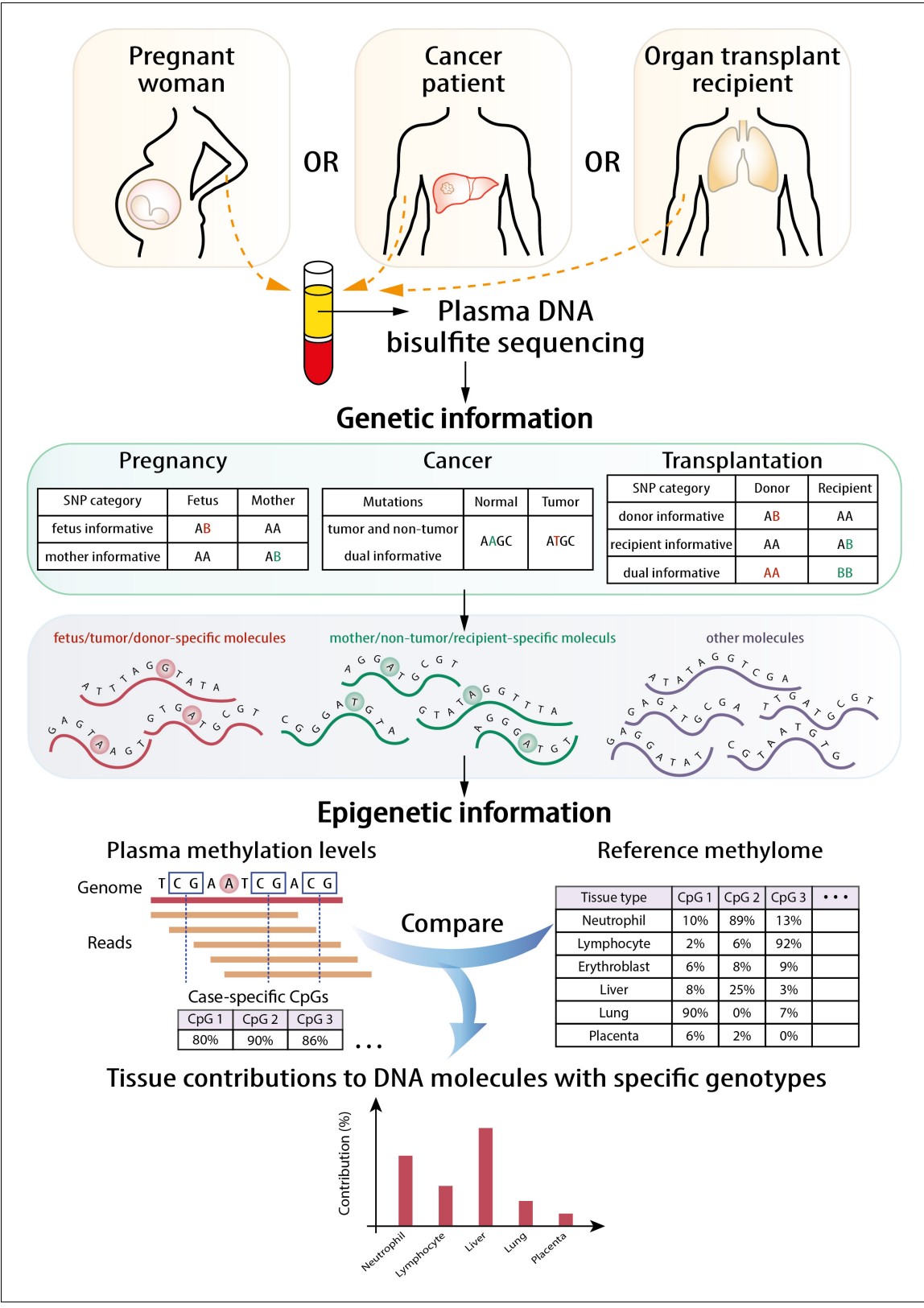

**Figure 1.** Schematic illustration of the principle of genetic-epigenetic tissue mapping (GETMap) analysis. The paired individuals (e.g., fetus/mother, organ donor/recipient, and tumor/normal tissue) are genotyped to identify single nucleotide polymorphism (SNP) alleles specific for one of them. After bisulfite sequencing, plasma DNA molecules carrying individual-specific alleles and at least one CpG site are identified. The plasma DNA methylome is

*Figure 1 continued*

compared with the methylation profiles of reference tissues to determine the tissue composition of the subset of plasma DNA molecules derived from a particular individual.

sites (range 16,796–138,284) were identified on the plasma DNA molecules carrying maternal-specific and fetal-specific alleles, respectively. For the plasma DNA molecules carrying fetal-specific alleles, the median deduced contribution from the placenta was 100% (*Figure 2A*). These results are compatible to the results of previous studies that fetal DNA in maternal plasma is derived from the placenta (*Alberry et al., 2007*; *Masuzaki et al., 2004*). For molecules carrying maternal-specific alleles, a median of 80% of DNA molecules were deduced to be derived from hematopoietic cells (i. e., neutrophils and lymphocytes) (*Figure 2B*). All cases showed no contribution from the placenta. For molecules carrying the shared alleles at SNPs where the mother was homozygous and the fetus was heterozygous, the deduced placental contribution showed a positive correlation with the fetal DNA fractions based on the ratio between the number of plasma DNA molecules carrying fetal-specific alleles and alleles shared by the mother and the fetus (*Figure 2C*).

## Deconvolution of DNA molecules carrying donor- and recipient-specific alleles following lung transplantation

We applied GETMap analysis to patients who had received lung transplantation and explored if the tissue composition would change over time. Forty samples from 11 patients were collected (*Table 2*). By comparing the SNP genotypes between the donor and recipient, we identified a median of 270,144 (range 254,846–344,024) donor-specific informative SNPs where the donor was heterozygous and the recipient was homozygous and a median of 270,285 (range 261,529–357,009)

**Table 1.** Results of deconvolution of bisulfite sequencing data from reference tissues for scenarios of (A) pregnancy, (B) lung transplantation, and (C) liver cancer.

The underlined numbers represent the percentage of contribution accurately assigned to the respective tissues by genetic-epigenetic tissue mapping (GETMap).

**(A)**

| | | Tissue contribution as determined by GETMap analysis | | | | |
|---|---|---|---|---|---|---|
| | | **Neutrophils** | **Lymphocytes** | **Liver** | **Lung** | **Placenta** |
| Reference tissue used for the simulation | Neutrophils | <u>96.78</u> | 2.01 | 0.59 | 0.33 | 0.29 |
| | Lymphocytes | 0.52 | <u>98.30</u> | 0.41 | 0.20 | 0.58 |
| | Liver | 0.31 | 0.64 | <u>98.36</u> | 0.27 | 0.42 |
| | Lung | 0.24 | 0.66 | 0.35 | <u>98.36</u> | 0.39 |
| | Placenta | 0.13 | 0.05 | 0.00 | 0.09 | <u>99.73</u> |

**(B)**

| | | Tissue contribution as determined by GETMap analysis | | | | |
|---|---|---|---|---|---|---|
| | | **Neutrophils** | **Lymphocytes** | **Liver** | **Lung** | **Placenta** |
| Reference tissue used for the simulation | Neutrophils | <u>98.21</u> | 0.77 | 0.42 | 0.43 | 0.17 |
| | Lymphocytes | 0.48 | <u>98.70</u> | 0.20 | 0.31 | 0.31 |
| | Liver | 0.32 | 0.19 | <u>99.25</u> | 0.11 | 0.13 |
| | Lung | 0.21 | 0.09 | 0.22 | <u>99.39</u> | 0.09 |
| | Placenta | 0.00 | 0.09 | 0.08 | 0.05 | <u>99.78</u> |

**(C)**

| | | Tissue contribution as determined by GETMap analysis | | | | |
|---|---|---|---|---|---|---|
| | | **Neutrophils** | **Lymphocytes** | **Liver** | **Lung** | **Placenta** |
| Reference tissue used for the simulation | Neutrophils | <u>96.08</u> | 2.23 | 0.32 | 0.37 | 1.00 |
| | Lymphocytes | 0.94 | <u>95.46</u> | 0.79 | 2.06 | 0.75 |
| | Liver | 0.50 | 0.44 | <u>96.67</u> | 1.48 | 0.91 |
| | Lung | 0.90 | 1.71 | 0.80 | <u>96.08</u> | 0.51 |
| | Placenta | 0.49 | 0.13 | 0.77 | 0.34 | <u>98.27</u> |

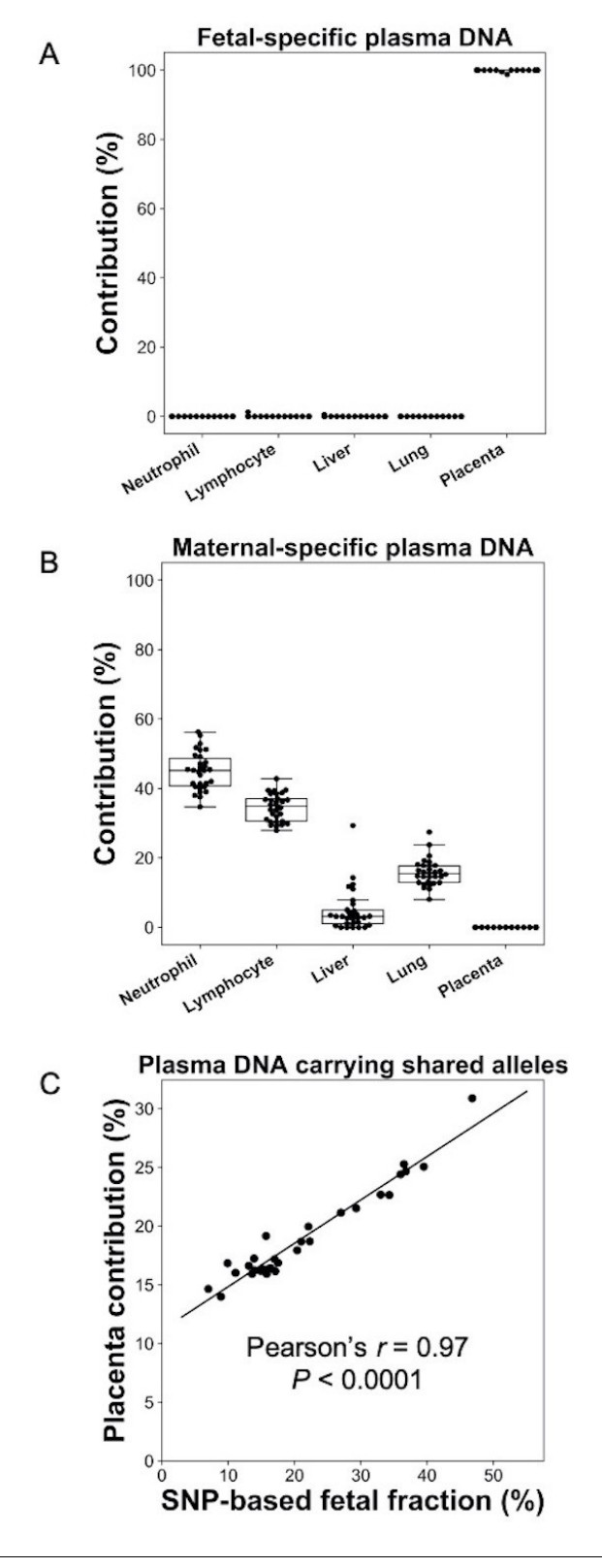

**Figure 2.** Percentage contributions of different cell types to maternal plasma DNA carrying (**A**) fetal-specific alleles and (**B**) maternal-specific alleles in 30 pregnant women. (**C**) Correlation between percentage contribution of the placenta to maternal plasma DNA molecules carrying alleles shared by the fetus and mother and single nucleotide polymorphism (SNP)-based fetal DNA fraction.

**Table 2.** The demographic profiles of lung transplant recipients.

| Case number | Recipient age | Recipient gender | Donor age | Donor gender | Diagnosis for transplant | Single/ double lung | Cause of death | Time of sample collection post-transplant |
|---|---|---|---|---|---|---|---|---|
| 1 | 34 | M | 32 | M | Cystic fibrosis | Double | Alive | 72 hr |
| 2 | 59 | F | 27 | F | Interstitial lung disease | Double | Alive | 72 hr |
| 3 | 53 | M | 20 | M | Interstitial lung disease | Double | Alive | 72 hr |
| 4 | 63 | M | 16 | F | Interstitial lung disease | Double | Alive | 72 hr, 6 dy |
| 5 | 55 | F | 36 | F | Interstitial lung disease | Double | Alive | 72 hr, 7 dy |
| 6 | 66 | M | 48 | F | Interstitial lung disease | Single | Alive | 72 hr, 4 wk |
| 7 | 66 | F | 18 | M | Chronic obstructive pulmonary disease | Single | Alive | 72 hr, 7 dy, 5 wk, 20 wk, 25 wk, 157 wk |
| 8 | 32 | F | 39 | M | Cystic fibrosis | Double | Alive | 72 hr, 7 dy, 8 wk, 38 wk, 77 wk, 129 wk |
| 9 | 67 | F | 53 | F | Sarcoidosis | Double | Respiratory failure | 72 hr, 7 dy, 6 wk, 13 wk, 22 wk |
| 10 | 44 | M | 35 | F | Retransplant | Double | Alive | 72 hr, 7 dy, 10 dy, 4 wk, 14 wk, 25 wk, 103 wk |
| 11 | 67 | F | 32 | M | Pulmonary arterial hypertension | Single | Alive | 72 hr, 7 dy, 5 wk, 15 wk, 26 wk, 61 wk, 104 wk |

*Samples collected when the patient was having a rejection episode were underlined.

recipient-specific informative SNPs where the donor was homozygous and the recipient was heterozygous. In addition, a median of 81,957 (range 77,196–133,422) dual informative SNPs where both the donor and recipient were homozygous but for different alleles were identified. After bisulfite sequencing of the plasma DNA, a median of 327 million uniquely mapped reads (range 32–481 million) were obtained for each case. A median of 920,830 (range 141,065–1,329,292) and 141,794 (range 12,700–529,211) CpG sites were identified on the plasma DNA molecules carrying recipient- and donor-specific alleles, respectively.

For each subject, the first sample was collected at 72 hr after the transplantation. We performed the GETMap analysis on donor-derived DNA molecules for each sample collected at 72 hr post-transplant (*Figure 3A*). The median contribution from the lung to the donor-derived DNA was only 17%. Surprisingly, a substantial proportion of the DNA molecules carrying the donor-specific alleles were contributed from the hematopoietic cells. The median contribution from the neutrophils and lymphocytes combined was 78%. The median deduced contribution from all other tissues was 5% in total.

We studied the changes in the lung DNA proportions in the donor-derived plasma DNA molecules with time after transplantation. We categorized the samples based on the time of sample collection post-transplant: within 72 hr; in-between 72 hr, 7 days, 10 weeks, and 50 weeks; and beyond 50 weeks. The 40 samples were thus classified into five categories that included 11, 7, 7, 9, and 6 samples, respectively. The median fractional concentrations of donor-derived DNA were 16%, 6%, 2%, 1%, and 2% for these categories, respectively (*Figure 3B*). The median contributions from the lung to the donor-derived DNA were 17%, 34%, 59%, 51%, 66% for samples in these categories, respectively (*Figure 3C*). These data showed that the lung DNA proportions in donor-derived DNA increased with time after transplantation. In contrast, the median contributions from the hematopoietic cells decreased with time, that is, 78%, 56%, 27%, 41%, and 21% for samples in the five categories, respectively. For the plasma DNA molecules carrying the recipient-specific alleles, we observed the hematopoietic cells as the key contributors. For samples in the five categories, the median contributions of hematopoietic cells were 83%, 86%, 89%, 94%, and 84%, respectively (*Figure 4*).

We further explored if the fractional contribution of the lung to the donor-specific DNA would be useful for the detection of graft rejection. As all the rejection episodes occurred after 7 days, only samples collected after 7 days were used for this analysis. The median donor-derived DNA fractions were 3% for the samples collected during rejection episodes and 1% for those collected during remission (p-value=0.22, Mann-Whitney rank-sum test, *Figure 3B*). The median lung contributions

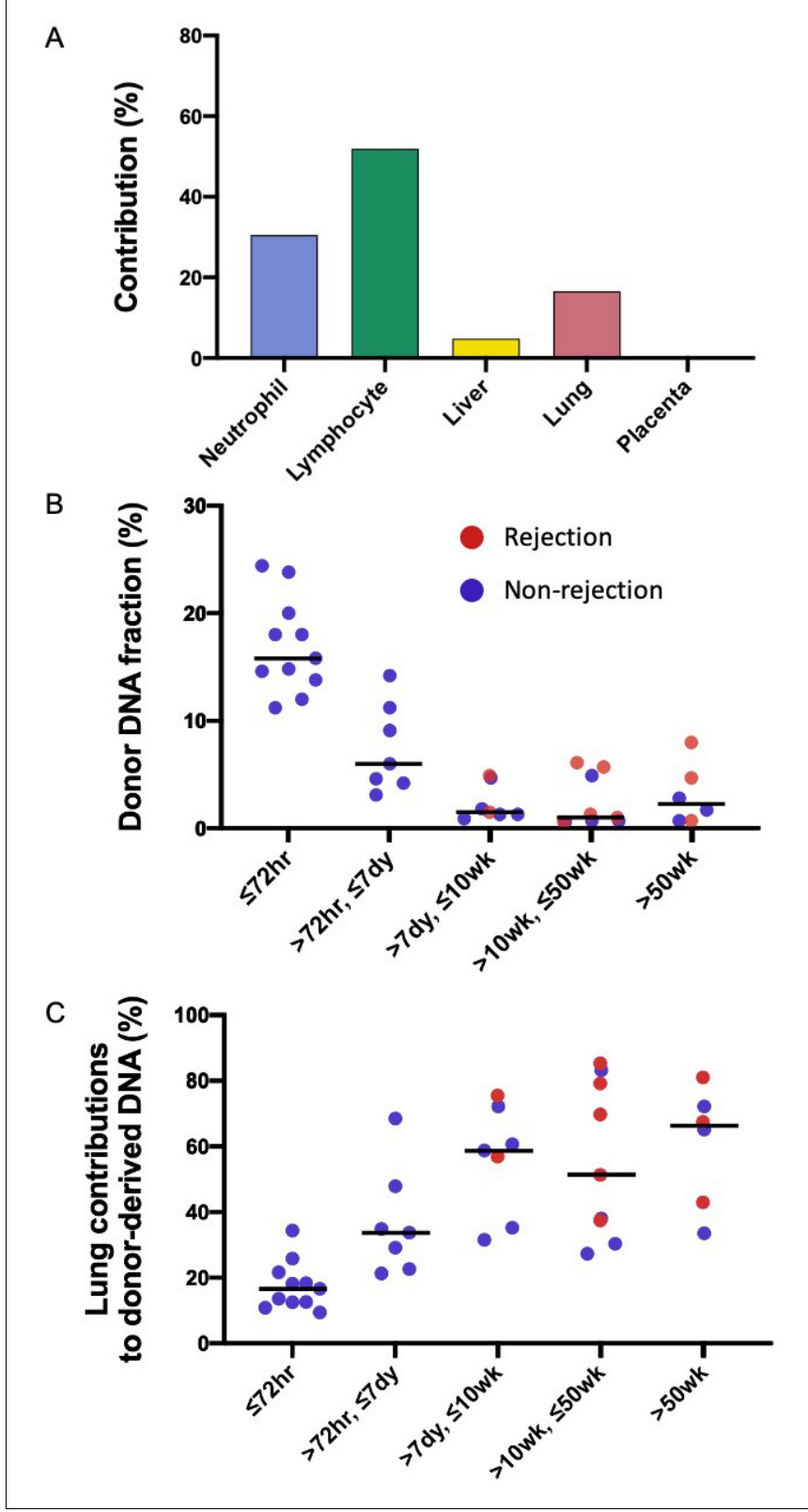

**Figure 3.** Genetic-epigenetic tissue mapping (GETMap) analysis on donor-derived plasma DNA molecules in lung-transplant recipients. (A) The median percentage contributions of different cell types to plasma DNA carrying donor-specific alleles in patients with lung transplantation at 72 hr post-transplant. (B) Fractional concentrations of

*Figure 3 continued on next page*

donor-derived DNA and (C) percentage contributions of the lung to plasma DNA carrying donor-specific alleles in patients with lung transplantation.

were 69% and 48% for these two groups of samples, respectively (p-value=0.09, Mann-Whitney rank-sum test, *Figure 3C*).

## Deconvolution of plasma DNA molecules carrying mutant identified in tumor tissues

We then explored if GETMap analysis could reveal the tissue origin of ctDNA in two HCC patients. The two patients were denoted as HCC 1and HCC 2, respectively. In the initial analysis, we first identified the cancer-specific mutations by analyzing the tumor tissues and the buffy coat of the patients. A total of 30,383 and 6996 tumor-specific single nucleotide mutations were identified from HCC 1 and HCC 2, respectively. After bisulfite sequencing of plasma DNA, 245 and 188 million uniquely mapped reads were obtained for the two patients, respectively. The numbers of plasma DNA molecules carrying the mutant alleles were 29,868 and 5090, and these molecules covered 18,193 and 4076 CpG sites, respectively. Tissue contributions of these tumor-derived plasma DNA molecules were deduced by GETMap analysis (*Figure 5*). The liver was deduced to be the key contributor with 90% (HCC 1) and 87% (HCC 2). A small contribution of 10% (HCC 1) and 13% (HCC 2) was from the placenta. The numbers of molecules carrying the wildtype alleles were 153,238 and 26,792, containing 35,883 and 8156 CpG sites, respectively. The contribution of the hematopoietic cells was deduced to be 48% (HCC 1) and 53% (HCC 2) whereas the liver contributed 32% (HCC 1) and 23% (HCC 2).

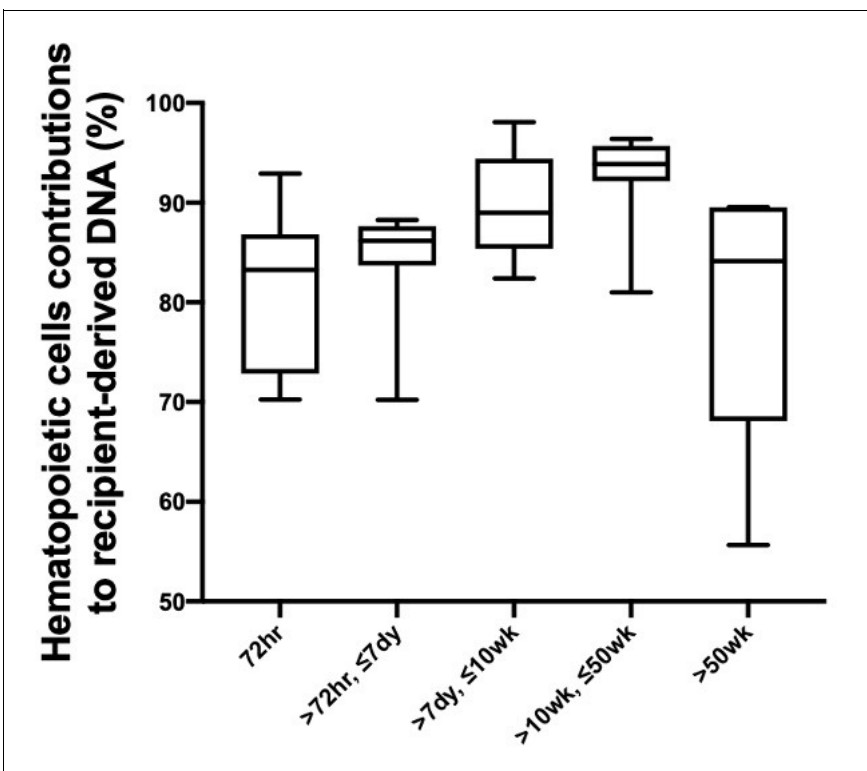

**Figure 4.** Percentage contributions of hematopoietic cells to the plasma DNA carrying recipient-specific alleles in patients with lung transplantation.

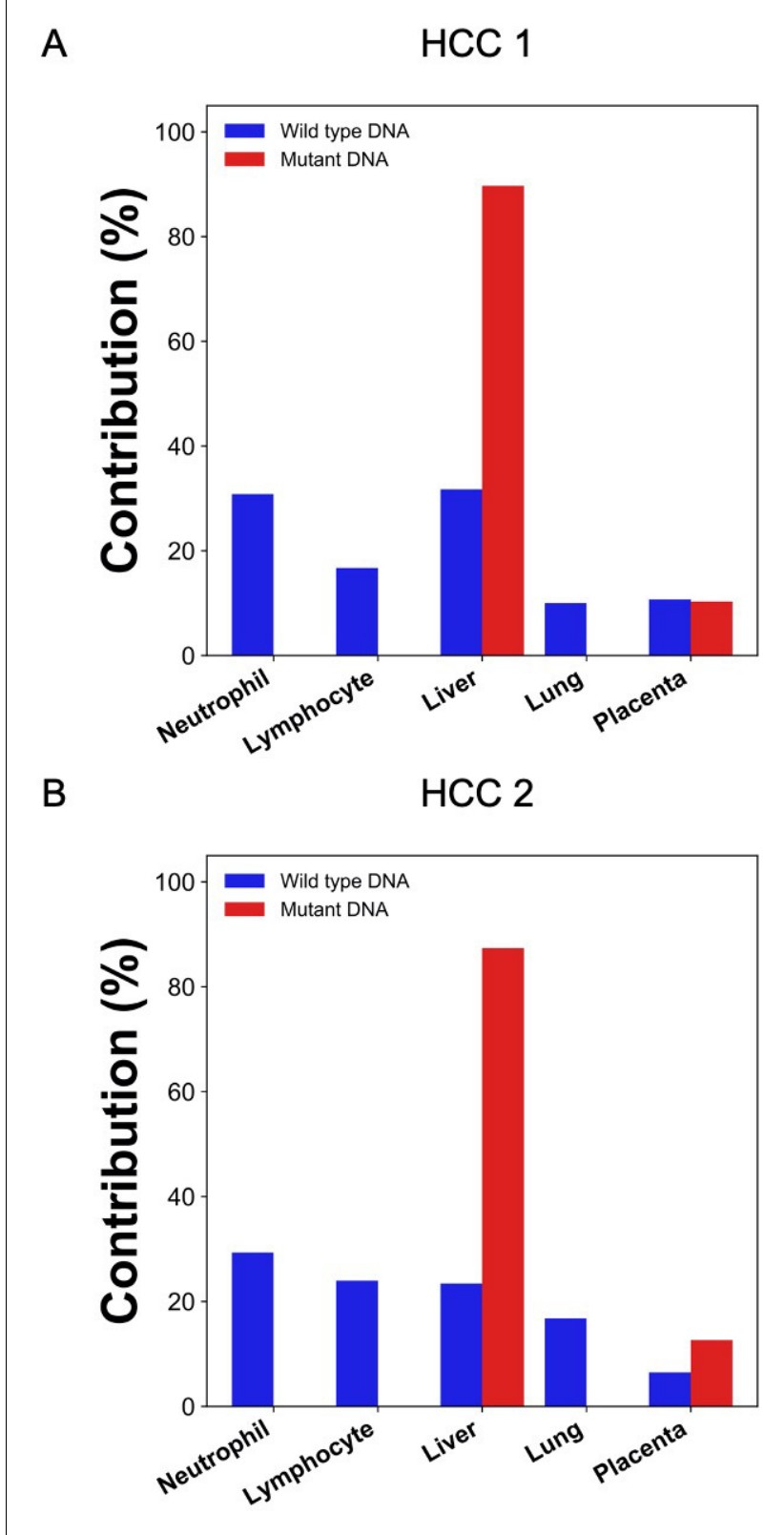

**Figure 5.** Percentage contributions of different tissues to plasma DNA with tumor-specific and wildtype alleles in two hepatocellular cancer (HCC) patients. The tumor-specific mutations were deduced from the tumor tissues.

## Deconvolution of DNA carrying mutations directly derived from plasma

In the scenario of cancer screening using a universal tumor marker based on plasma DNA analysis, the tumor tissue would not be available for mutation analysis. Hence, we further explored if the cancer mutations can be directly derived from plasma DNA analysis. To obtain the mutation information directly from the plasma DNA, we sequenced the buffy coat and plasma DNA without bisulfite conversion. The sequencing depth for the plasma DNA were 50x and 61x haploid genome coverage and those for the buffy coat DNA were 53x and 55x in HCC 1 and HCC 2, respectively. Single nucleotides variations present in the plasma for more than a threshold number of occasions but not in the buffy coat were identified as candidate mutations (see details in the 'Materials and methods'). The numbers of candidate mutations identified were 10,864 and 3446 for the two HCC patients. GETMap analysis was then performed using the plasma DNA bisulfite sequencing data. The numbers of plasma DNA molecules carrying the cancer mutations were 16,200 and 4112, and covered 12,887 and 2991 CpG sites, respectively. For molecules carrying mutations, the contributions from the liver were estimated to be 69% (HCC 1) and 95% (HCC 2) (*Figure 6*). The placenta contributed the remaining proportion of 31% (HCC 1) and 5% (HCC 2). For molecules carrying wildtype alleles, hematopoietic cells, including neutrophils and lymphocytes, contributed a total of 51% (HCC 1) and 27% (HCC 2).

## Deconvolution of plasma DNA for a pregnant woman with lymphoma

We previously reported the deconvolution results of total plasma DNA for a pregnant woman who was diagnosed as having follicular lymphoma during early pregnancy (*Sun et al., 2015*). In the current study, we explored if GETMap analysis could determine the tissue composition of the fetal- and cancer-derived DNA independently. We sequenced the lymphoma tissue, as well as the normal cells harvested from buccal swab and post-treatment buffy coat. As the pregnancy was terminated at time of the diagnosis of cancer, no placental tissue was collected. Hence, we deduced the fetal genotypes directly from the plasma DNA. Based on the non-bisulfite sequencing results of the plasma DNA and normal cells, 254,540 variants were identified in the plasma DNA. The algorithm for classifying these variants into fetal-specific alleles and cancer mutations is shown in *Figure 7*. We reasoned that variants overlapping with the common variations in the dbSNP Build 135 database were more likely derived from the fetus whereas those not overlapping with the database were more likely to come from the tumor. For the 13,546 variants that did not overlap with dbSNP database, 2641 were detected in three or more sequence reads of the tumor tissues. These variants are regarded as tumor mutations for GETMap analysis. For the 240,994 variants overlapping with the dbSNP database, 231,552 were completely absent in the tumor tissue. These variants were likely derived from the fetus and are regarded as fetal-specific alleles for the GETMap analysis. The allele frequencies for the fetal-specific SNPs and tumor-specific mutations in plasma were normally distributed and peaked at 6% and 20%, respectively (*Figure 8*).

After bisulfite sequencing of plasma DNA, we obtained 700 million uniquely mapped reads. We identified DNA molecules carrying the tumor-specific mutant alleles, wildtype alleles, fetal-specific alleles, and the alleles shared by the fetus and the mother. The GETMap analysis was performed on each set of plasma DNA molecules to deduce their tissue composition. The numbers of CpG sites covered by the DNA molecules carrying the mutant and wildtype alleles were 4781 and 6660, respectively. For the molecules carrying tumor mutations, it was deduced that 100% was from lymphocytes (*Figure 9A*). For molecules carrying the wildtype alleles, the deduced contribution from neutrophils, lymphocytes, liver, lung, and placenta were 29%, 46%, 13%, 2%, and 11%, respectively. For DNA molecules carrying the fetal-specific, the deduced contribution from the placenta was 95% (*Figure 9B*). For those carrying alleles shared by the mother and fetus, the deduced contribution from neutrophils, lymphocytes, liver, lung, and placenta were 23%, 48%, 11%, 14%, and 5%, respectively.

## Discussion

In this study, we developed GETMap analysis to determine the tissue origin of plasma DNA molecules carrying genetic variants. In this method, we first identified a subset of plasma DNA molecules carrying specific alleles. Then, by comparing the methylation status of these molecules and the

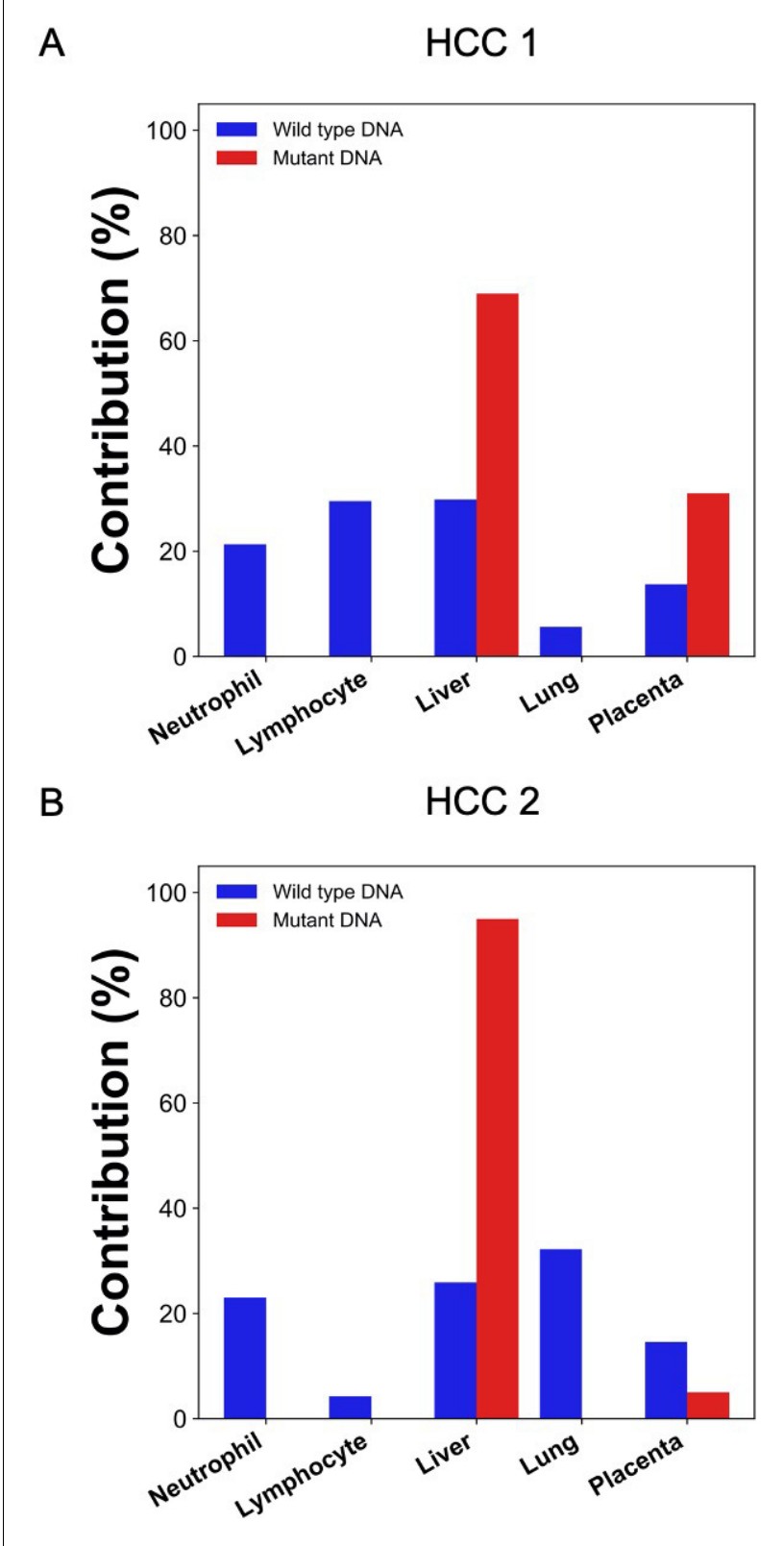

**Figure 6.** Percentage contributions of different tissues to plasma DNA with tumor-specific and wildtype alleles in two hepatocellular cancer (HCC) patients. The tumor-specific mutations were deduced directly from the plasma.

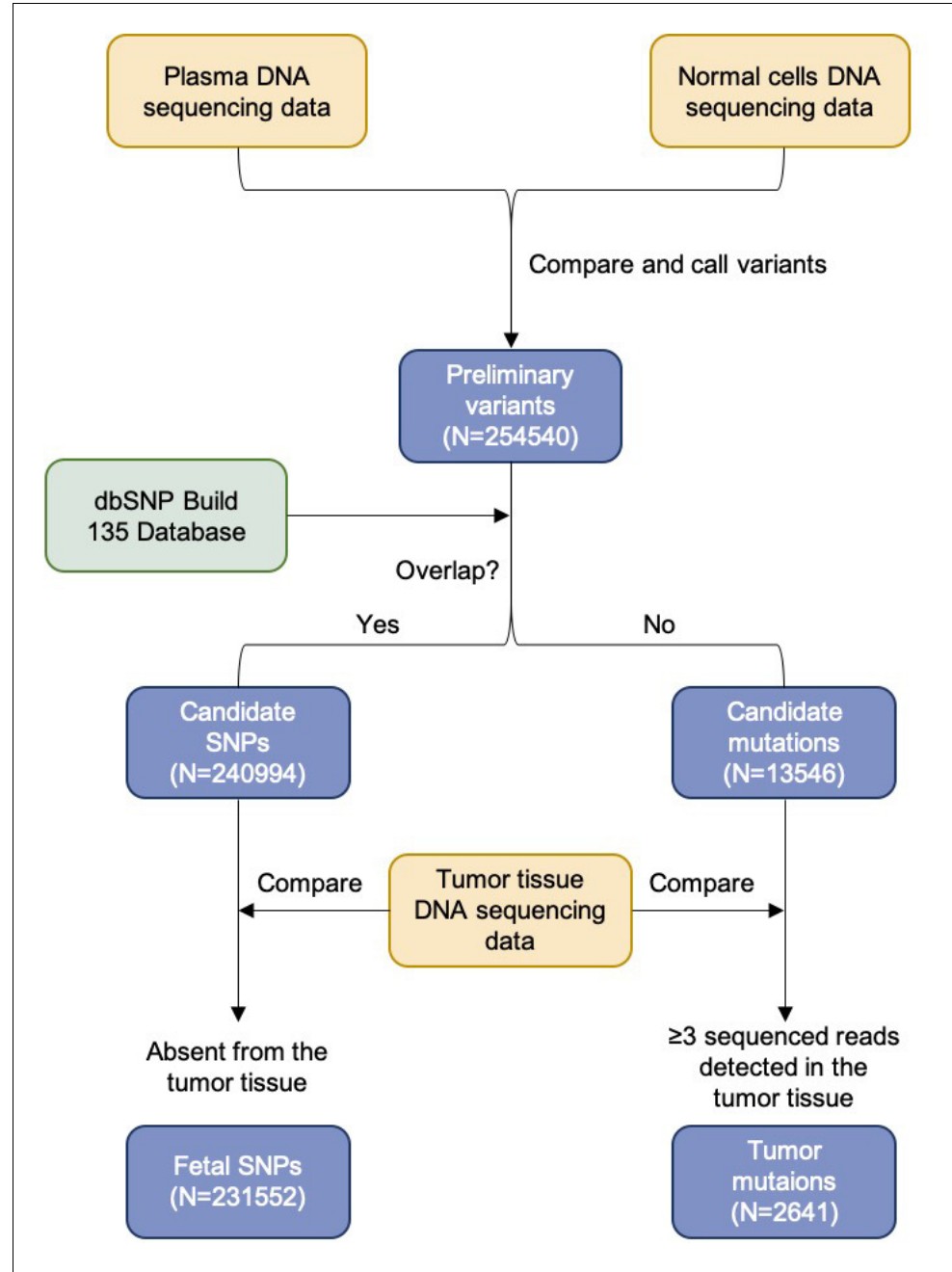

**Figure 7.** Flowchart of the steps for identifying the fetal-specific alleles and cancer mutations in the pregnant woman with lymphoma.

methylation profiles of the candidate tissue organs, we could determine the tissue composition of the DNA molecules. In the first part of the study, we used the pregnancy model to validate the GET-Map analysis. The plasma DNA molecules carrying the fetal-specific alleles were deduced to be 100% derived from the placenta. For the molecules carrying the alleles shared by the fetus and the mother, the percentage contribution from the placenta showed a positive linear relationship with the fractional concentration of fetal DNA based on SNP analysis. These results are consistent with the previous studies which showed that the fetal DNA in maternal plasma is indeed derived from the placenta. For the plasma DNA molecules carrying maternal-specific alleles, no contribution from the placenta was observed. A large proportion was derived from the hematopoietic cells, neutrophils, and lymphocytes, with a median total contribution of 80%. These figures are comparable to those

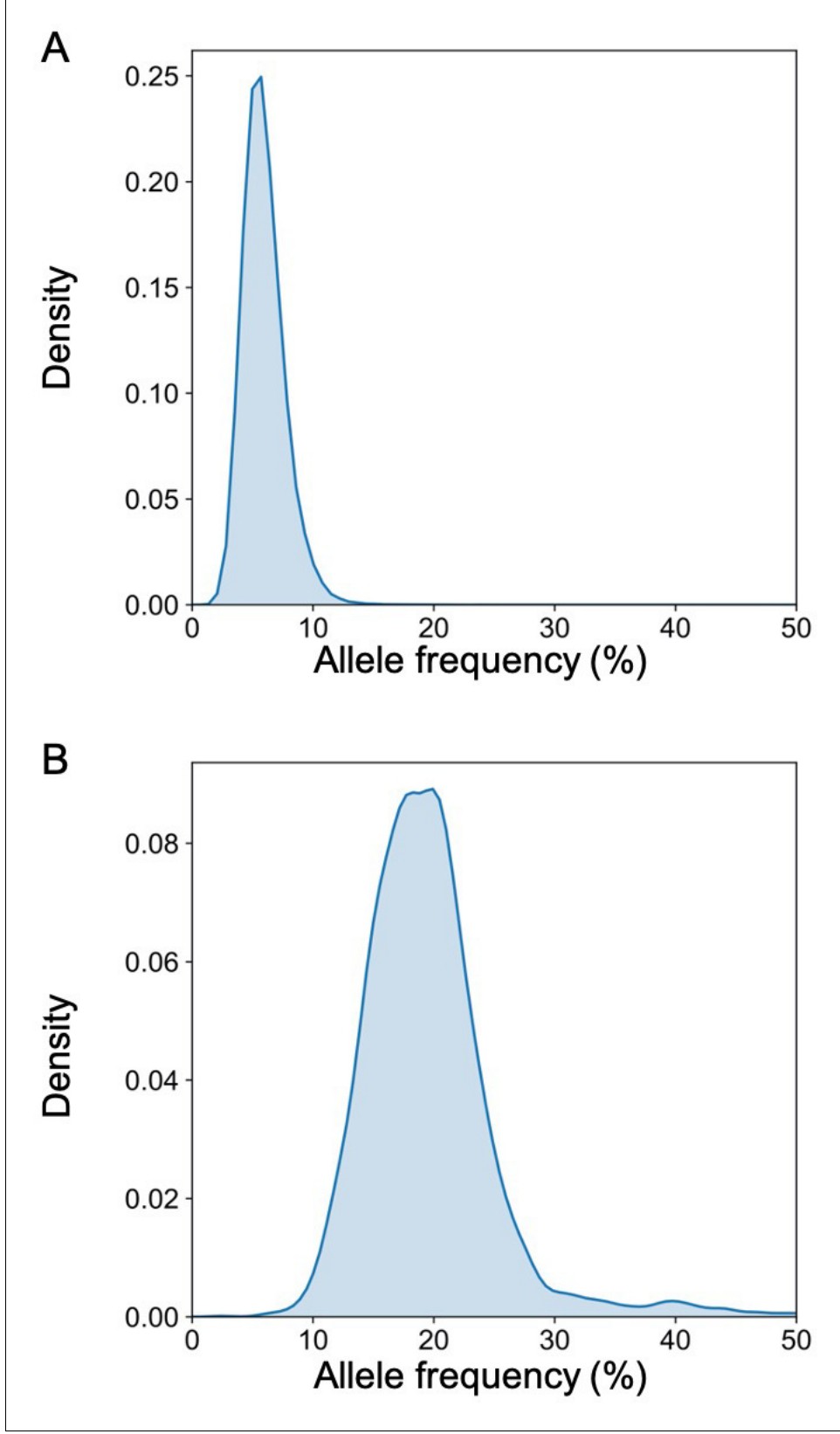

**Figure 8.** The distribution of the allele frequency of (A) the fetal-specific alleles and (B) the mutant alleles in the plasma of the pregnant woman with lymphoma.

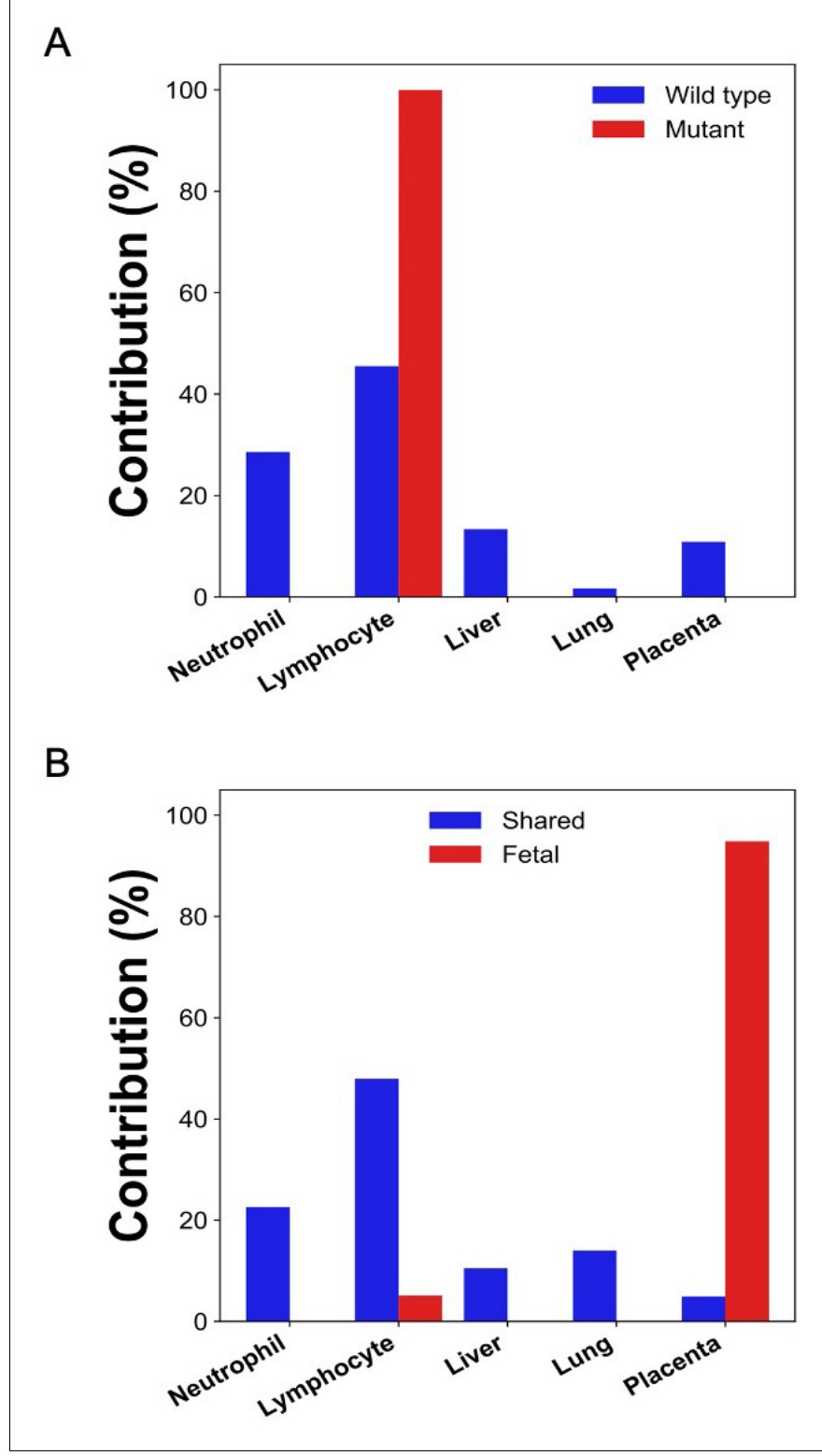

**Figure 9.** Percentage contributions of different tissues to (**A**) plasma DNA with tumor-specific and wildtype alleles, and (**B**) fetal-specific plasma DNA and DNA carrying the alleles shared by the fetus and the mother in a pregnant woman with lymphoma.

reported previously in healthy subjects (*Gai et al., 2018*; *Sun et al., 2015*). These results demonstrate the feasibility of determining the tissue contributions to the different genetic components of plasma DNA using GETMap analysis.

We then showed that, in patients who had received lung transplantation, a substantial proportion of donor-derived DNA was derived from the hematopoietic cells during the early post-transplant period. Previous studies have shown that a high level of DNA carrying donor genotypes would be present in the plasma of organ transplant recipients during the early post-transplant period even in the absence of any evidence of organ rejection (*De Vlaminck et al., 2015*). Hence, quantitative analysis for donor DNA in plasma cannot be used for reflecting transplant organ damage or rejection within 60 days of transplantation. The reason for this elevation in donor DNA was unclear. Using GETMap analysis, we determined the tissue composition of plasma DNA molecules carrying donor-specific alleles for samples collected at different time intervals after transplantation. Importantly, at 72 hr after transplantation, the median contribution from the lung was only 17% and a substantial contribution of 78% was from hematopoietic cells. This is likely due to the presence of residual blood cells in the transplanted organ and they could release DNA with donor genotypes into the circulation. The contribution of the lung gradually increases with time together with a parallel decline in the contribution of the hematopoietic cells. The median contribution of hematopoietic cells dropped to 21% after 50 weeks. The persistent contribution from the hematopoietic cells may be due to imprecision of measurement as the concentrations of donor DNA after 50 weeks were very low in patients without evidence of rejection. Alternatively, there may be persistence of donor hematopoietic cells in the body of the transplant recipient. In this regard, it has been shown that some immune cells resident in the donor tissue can be long-lived and self-renewing (*Gasteiger et al., 2015*). The lung fraction appeared to be higher for samples collected during graft rejection compared with those collected during remission. However, the difference did not reach statistical significance. Future studies with larger sample size would be useful to further explore this point.

We then investigated if GETMap analysis could be used to identify the tissue origin of plasma DNA derived from the tumor. Circulating DNA analysis has increasingly been used in the management of cancer patients, in particular for guiding the use of target therapy and monitoring disease progression (*Mok et al., 2017*; *Wan et al., 2020*; *Yung et al., 2009*). Recently, it has been demonstrated that the analysis for cancer-derived DNA in plasma is useful for the screening of cancers in asymptomatic individuals (*Chan et al., 2017*; *Lennon et al., 2020*). As genetic and methylation aberrations are present in almost all cancers, the detection of cancer-associated alterations in plasma DNA can potentially serve as a universal tumor marker for a wide variety of cancers. The capability of a tumor marker for picking up multiple types of cancers can greatly enhance the cost-effectiveness of a cancer screening program. However, the lack of tissue or organ specificity of these tests also poses practical challenges on the workup of subjects with positive test results. In the screening study by Lennon et al., subjects tested positive were further investigated with PET-CT to confirm and localize a possible tumor (*Lennon et al., 2020*). However, if the tissue origin and location can be obtained from the ctDNA analysis, more targeted investigation on the potentially affected organ may be performed. For example, a colonoscopy can be performed for individuals who are suspected of having colorectal cancers. This targeted investigation approach not only provides a more accurate assessment for cancers, it also reduces the radiation exposure of the tested positive subjects. Here, we used the GETMap analysis to determine the tissue origin of plasma DNA carrying cancer-associated mutations. First, we compared the sequencing results of the tumor tissues and the blood cells to identify the mutations in the tumor tissues of two HCC patients. In contrast to the pregnancy and transplantation models which used microarray for genotyping, we used whole-genome sequencing to identify the cancer-associated mutations as these mutations would not be covered by the whole-genome arrays. In our simulation analysis, the numbers of informative SNPs and mutations identified are shown to provide a median accuracy of 98.3%. After bisulfite sequencing of the plasma DNA, DNA molecules carrying the cancer-associated mutations were identified and their methylation profiles were used to deduce the contribution from different tissues. The liver was deduced to be the key contributor to these cancer-derived plasma DNA molecules with a contribution of 90% and 87% for the two male HCC patients. The remaining portion, that is, 10% and 13%, were attributed to placental contribution. The attribution of a small proportion of ctDNA to originate from the placenta may be due to the fact that global hypomethylation and hypermethylation of tumor suppressor genes are common features in both the placenta and tumor tissues (*Chan et al., 2013a*;

*Feinberg and Vogelstein, 1983*; *Lun et al., 2013*). Although this analysis suggests that GETMap analysis may be useful for revealing the tissue origin of ctDNA, the requirement of tumor tissues for mutation identification limits its practical application in cancer screening. To overcome this, we further attempted to identify cancer mutations directly from plasma DNA sequencing. In this regard, non-bisulfite sequencing for the plasma DNA and the blood cells of the cancer patients were performed. The single nucleotide variants present in the plasma DNA but not in the blood cells were regarded as cancer-associated mutations. GETMap analysis was performed on the plasma DNA molecules carrying these mutations using the bisulfite sequencing data. Despite a smaller number of cancer-associated mutations could be identified by directly sequencing plasma DNA compared with sequencing the tumor tissues, the liver was again correctly identified as the key contributor to these cancer-derived DNA molecules. These results suggest that the GETMap analysis could be useful in revealing the tissue origin and location of a concealed cancer in patients who are screened positive with a tumor marker that detects various types of cancers.

We further challenged GETMap analysis with a complex scenario where a woman developed lymphoma during pregnancy. Her plasma consisted of DNA derived from the lymphoma tissues, the fetus, and the normal cells. As fetal tissue was not available, fetal genotypes were deduced by sequencing plasma DNA, maternal blood cells/buccal cells, and tumor tissues. Sequence variants present in plasma that overlap with the dbSNP database but absent in the tumor tissues were regarded as fetal-specific alleles. Variants detected in plasma and tumor tissues, but not overlapping with the dbSNP database were regarded as tumor-specific. Plasma DNA molecules carrying these fetal-specific alleles were deduced to be predominantly (95%) derived from the placenta, whereas those carrying the tumor-specific alleles were solely from lymphocytes.

There has been increasing interest in the tissue composition circulating cell-free DNA. Methods based on analysis of DNA methylation (*Gai et al., 2018*; *Lehmann-Werman et al., 2016*; *Sun et al., 2015*), nucleosome footprint (*Snyder et al., 2016*; *Sun et al., 2019*), sequence motifs, end coordinates, and jaggedness (*Chan et al., 2016*; *Jiang et al., 2018*; *Jiang et al., 2020a*; *Jiang et al., 2020b*) have been developed. However, existing methods only allow the deconvolution of all the DNA as a single entity. In contrast, GETMap analysis can determine the tissue origin of subsets of plasma DNA that carry different genetic variations. The specific analysis of a particular component can enhance the signal-to-noise ratio and eliminate the variation caused by the difference in the concentrations of the target DNA, for example, DNA derived from the tumor. Furthermore, clonal hematopoiesis has been identified as one important source of false-positive results for liquid biopsy-based cancer screening tests. In this regard, GETMap would be useful for identifying the hematopoietic origin of the abnormal signal in such cases. Although the number of cases is relatively small in this proof-of-principle study, we have illustrated the potential applications in cancer detection, prenatal testing, and organ transplant monitoring. As the current format of this method is based on whole-genome bisulfite sequencing, identification of cytosine to thymine alteration is less efficient because bisulfite treatment would convert unmethylated cytosine to thymine. A targeted sequencing approach enriching for regions with mutation hotspots and differential methylation across different tissues can be developed to enhance the cost-effectiveness of this approach.

## Materials and methods

### Samples and processing

The project was approved by the Joint Chinese University of Hong Kong-Hospital Authority New Territories East Cluster Clinical Research Ethics Committee (approval reference number 2011.204). All participants provided written informed consent. Pregnant women and HCC patients were recruited from the Prince of Wales Hospital of Hong Kong. The pregnant woman with lymphoma was recruited from the Hong Kong Sanatorium and Hospital, Hong Kong. Lung transplant recipients were recruited from the National Institutes of Health (NIH) (iRIS reference number 363880). Plasma samples were collected longitudinally at one or several time points after transplantation. Venous blood samples were collected into EDTA-containing tubes and centrifuged at 1600 *g* for 10 min. The plasma portion was recentrifuged at 16,000 *g* to remove residual blood cells. DNA from plasma was extracted with the QIAamp Circulating Nucleic Acid Kit (Qiagen).

## Identification of tumor-specific mutations in HCC patients

We prepared libraries using DNA extracted from the tumor tissue and buffy coat with the TruSeq Nano DNA Library Prep Kit (Illumina). Paired-end (2 × 75 bp) sequencing was performed on the HiSeq4000 system (Illumina). Sequencing data were aligned to the human reference genome using the Burrows-Wheeler Aligner (*Li and Durbin, 2010*). We compared the data of tumor tissue with that of buffy coat to call the tumor-specific mutations using the Genome Analysis Toolkit (version 4.1.2.0) (*McKenna et al., 2010*).

To call the tumor-specific mutations directly from the plasma, DNA isolated from the plasma was submitted to library preparation and sequencing. The sequencing data of plasma DNA were then compared with that of the buffy coat to identify the tumor-specific mutations. Single nucleotides variations observed in plasma for more than a threshold number of occasions but not in the buffy coat were identified as candidate mutations. The threshold was based on the total number of sequenced reads covering the variant's nucleotide position as described in our previous study (*Chan et al., 2016*). In addition, the sequencing reads covering these candidate mutations were realigned to the reference human genome using a second alignment software which could reduce the number of false-positive results caused by alignment errors as described previously (*Chan et al., 2016*).

## Identification of tumor-specific mutations and fetal-specific SNPs in the pregnant women with lymphoma

The DNA extracted from the maternal plasma, tumor cells, and normal cells were submitted to library preparation using either the KAPA HTP Library Preparation Kit (Kapa Biosystems) or the Tru-Seq Nano DNA Library Prep Kit (Illumina) following the manufacturer's instructions. The 2 × 75 (paired-end mode) cycles of sequencing were performed using the Illumina platforms, including the HiSeq and NextSeq. To call the plasma-specific variants, we compared the sequencing data of DNA extracted from the maternal plasma with that from the normal cells using the dynamic cutoff algorithm as described previously (*Chan et al., 2016*). We used the biallelic SNPs downloaded from the dbSNP database (Build 135) to classify the plasma-specific variants. For plasma-specific variants within the dbSNP database, we further filtered out the variants that present in the tumor tissue to obtain the fetal-specific SNPs. For the non-dbSNP variants, the single nucleotide variants observed in at least three molecules from the tumor tissue sequencing data were remained as tumor-specific variants. The bioinformatic pipeline for filtering these mutations was written in Python script.

## Microarray-based genotyping

Pre-transplant blood samples were collected from the donor and recipient. Genomic DNA was extracted from whole blood with the DNeasy Blood and Tissue Kit (Qiagen) and amplified with REPLI-g Mini Kit (Qiagen). For the pregnant case, genomic DNA of the mother and fetus were extracted from maternal buffy coat and fetal placenta tissue with the QIAamp DNA Mini Kit (Qiagen). Genotyping was performed on Illumina whole-genome arrays (HumanOmni2.5 or Human-Omni1) following the manufacturer's protocol (*De Vlaminck et al., 2014*).

## Bisulfite-treated DNA libraries preparation and sequencing analysis

Libraries were prepared from plasma DNA with the TruSeq Nano DNA Library Prep Kit (Illumina). DNA libraries were subjected to two rounds of bisulfite modification with the EpiTect Bisulfite Kit (Qiagen) following by 12 cycles of PCR amplification. Bisulfite-treated libraries were sequenced in paired-end mode (2 × 75 bp) on a HiSeq 4000 system (Illumina). The sequencing reads were trimmed to remove adapter sequences and low-quality bases (i.e., quality score <5). The trimmed reads were aligned to the human reference genome build hg19 with Methy-Pipe (*Jiang et al., 2014*).

## GETMap analysis

The reference methylomes included the whole-genome bisulfite sequencing data of five different tissues, including neutrophils, lymphocytes (combining B and T lymphocytes), liver, and lung from the BLUEPRINT Project (*Martens and Stunnenberg, 2013*), Roadmap Epigenomics (*Roadmap Epigenomics Consortium et al., 2015*), ENCODE (*Davis et al., 2018*), and GEO (*Barrett et al., 2013*). In addition, bisulfite sequencing data of two placenta tissues generated by

our group were used as tissue-specific methylomes. The sequencing reads were aligned to the human reference genome build hg19 with bwa-meth (https://github.com/brentp/bwa-meth). After alignment, the methylation levels for 28,217,006 CpG sites across five types of tissues were determined. CpG sites fulfilling the following criteria were used for the analysis: (i) in the five reference tissues, the difference between the highest and lowest methylation levels was greater than 25% and (ii) after removing either tissue with the highest or the lowest methylation level, the coefficient of variation of methylation level across the remaining reference tissues was less than 0.3. We retrieved the methylation levels of different tissues across the set of CpG sites covered by the set of DNA molecules carrying the genetic variants. The measured CpG methylation levels of DNA molecules were recorded in a vector ($X$) and the retrieved reference methylation levels across different tissues were recorded in a matrix ($M$). The proportional contributions ($P$) from different tissues to donor- or recipient-specific DNA molecules were deduced by quadratic programming:

$$\bar{X}_i = \sum_k (p_k \times M_{ik}),$$

where $\bar{X}_i$ represents the methylation density of a CpG site $i$ in the DNA mixture; $p_k$ represents the proportional contribution of cell type $k$ to the DNA mixture; $M_{ik}$ represents the methylation density of the CpG site $i$ in the cell type $k$. When the number of sites is the same or larger than the number of organs, the values of individual $p_k$ could be determined.

The aggregated contribution of all cell types would be constrained to be 100%:

$$\sum_k p_k = 100\%$$

Furthermore, all the organs' contributions would be required to be non-negative:

$$p_k \geq 0, \, \forall \, k$$

The GETMap deconvolution analysis was performed with a program written in Python (http://www.python.org/).

## Sample information

The information of all the samples analyzed in this study, including sequencing depth, number of informative SNPs, number of informative sequencing fragments, number of informative CpG sites, and number of CpG sites used for deconvolution, are provided in *Supplementary file 1*.

## Acknowledgements

This work was supported by the Research Grants Council of the Hong Kong SAR Government under the theme-based research scheme (T12-403/15 N and T12-401/16 W), a collaborative research agreement from Grail and the Vice Chancellor's One-Off Discretionary Fund of The Chinese University of Hong Kong (VCF2014021). YMD Lo is supported by an endowed chair from the Li Ka Shing Foundation.

## Additional information

### Competing interests

YM Dennis Lo: Reviewing editor, eLife. Holds equities in DRA, Take2 and Grail. Serves as a scientific cofounder and consultant of Grail. Receives research funding from Grail. Receives royalties from Grail, Illumina, Sequenom, DRA, Take2 and Xcelom. Filed a patent application (US15/214,998). Peiyong Jiang: Holds equities in Grail. Serves as a director of KingMed Future. Received patent royalties from Grail, Illumina, Sequenom, DRA, Take2 and Xcelom. Filed a patent application (US15/214,998). Rossa WK Chiu: Holds equities in DRA, Take2 and Grail. Is a consultant to Grail and Illumina. Receives research funding from Grail. Receives royalties from Grail, Illumina, Sequenom, DRA, Take2 and Xcelom. Filed a patent application (US15/214,998). KC Allen Chan: Holds equities in DRA, Take2 and Grail. Is a consultant to and receives research funding from Grail. Receives royalties

from Grail, Illumina, Sequenom, DRA, Take2 and Xcelom. Filed a patent application (US15/214,998). The other authors declare that no competing interests exist.

## Funding

| Funder | Grant reference number | Author |
|---|---|---|
| Research Grants Council, University Grants Committee | Theme-based research scheme T12-403/15-N | Rossa WK Chiu<br>KC Allen Chan<br>YM Dennis Lo |
| Research Grants Council, University Grants Committee | Theme-based research scheme T12-401/16-W | Rossa WK Chiu<br>KC Allen Chan<br>YM Dennis Lo |
| Chinese University of Hong Kong | VCF2014021 | Rossa WK Chiu<br>KC Allen Chan<br>YM Dennis Lo |
| Grail | Collaborative research agreement | Rossa WK Chiu<br>KC Allen Chan<br>YM Dennis Lo |
| Li Ka Shing Foundation | | YM Dennis Lo |

The funders had no role in study design, data collection and interpretation, or the decision to submit the work for publication.

## Author contributions

Wanxia Gai, Formal analysis, Investigation, Visualization, Methodology, Writing - original draft; Ze Zhou, Data curation, Software, Methodology, Writing - review and editing; Sean Agbor-Enoh, Data curation, Investigation, Writing - review and editing; Xiaodan Fan, Sheng Lian, Investigation, Methodology, Writing - review and editing; Peiyong Jiang, Data curation, Investigation, Methodology, Writing - review and editing; Suk Hang Cheng, Formal analysis, Investigation, Methodology, Writing - review and editing; John Wong, Moon Kyoo Jang, Yanqin Yang, Raymond HS Liang, Wai Kong Chan, Edmond SK Ma, Tak Y Leung, Hannah Valantine, Investigation, Writing - review and editing; Stephen L Chan, Investigation; Rossa WK Chiu, Conceptualization, Data curation, Formal analysis, Funding acquisition, Investigation, Methodology, Writing - original draft, Project administration; KC Allen Chan, Conceptualization, Data curation, Formal analysis, Supervision, Investigation, Methodology, Writing - original draft, Project administration; YM Dennis Lo, Conceptualization, Resources, Formal analysis, Supervision, Funding acquisition, Validation, Investigation, Methodology, Writing - original draft, Project administration, Writing - review and editing

## Author ORCIDs

Xiaodan Fan (iD) http://orcid.org/0000-0002-2744-9030
KC Allen Chan (iD) https://orcid.org/0000-0003-1780-1691
YM Dennis Lo (iD) https://orcid.org/0000-0001-8746-0293

## Ethics

Human subjects: The project was approved by the Joint Chinese University of Hong Kong-Hospital Authority New Territories East Cluster Clinical Research Ethics Committee (approval reference number 2011.204). All participants provided written informed consent.

## Decision letter and Author response

Decision letter https://doi.org/10.7554/eLife.64356.sa1
Author response https://doi.org/10.7554/eLife.64356.sa2

## Additional files

### Supplementary files

• Supplementary file 1. The information of all the samples analyzed in this study, including sequencing depth, number of informative single nucleotide polymorphisms (SNPs), number of informative sequencing fragments, number of informative CpG sites, and number of CpG sites used for deconvolution.

• Transparent reporting form

### Data availability

Sequencing data have been deposited in EGA under the accession code EGAS00001004788.

The following dataset was generated:

| Author(s) | Year | Dataset title | Dataset URL | Database and Identifier |
|---|---|---|---|---|
| Gai W, Zhou Z, Jiang P, Cheng SH, Chiu RWK, Chan KCA, Lo YMD | 2021 | Methylation analysis for plasma DNA of patients with organ transplantation | https://www.ebi.ac.uk/ega/studies/EGAS00001004788 | The European Genome-phenome Archive, EGAS00001004788 |

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
