## [Decision Letter]

**Acceptance summary:**

Based on whole-genome bisulfite sequencing and tissue-specific methylation patterns, the authors reported a method to deconvolute the tissue origin of cell-free DNA in plasma. This approach allows mutation and methylation analyses on the same sequence read. Both reviewers find the described method original and of potential use in clinical settings.

**Decision letter after peer review:**

Thank you for submitting your article "Applications of genetic-epigenetic tissue mapping for plasma DNA in prenatal testing, transplantation and oncology" for consideration by *eLife*. Your article has been reviewed by two peer reviewers, and the evaluation has been overseen by a Reviewing Editor and Mone Zaidi as the Senior Editor. The reviewers have opted to remain anonymous.

Essential Revisions:

1) Please provide the following methodological details to allow an interested reader to repeat the analysis: (a) sequencing depth; (b) description of the methylation sites used for deconvolution, including selection criteria and number; (c) assessment of confidence and assignment of tissue origins. Please also describe Issues, if any, with determining SNPs after bisulfite conversion.

2) The reference data for tissue-specific methylomes should be better described, including a demonstration of the specificity of tissue mapping.

3) The authors generally obtained genotyping data from the whole-genome arrays lacking low-frequency and rare genotypes. The authors should consider improving the performance of the method by increasing the number of informative genotypes using genotype imputation or whole-exome or whole-genome sequencing. This limitation could be addressed in the Discussion.

---

## [Author Response]

Essential Revisions:1) Please provide the following methodological details to allow an interested reader to repeat the analysis: (a) sequencing depth; (b) description of the methylation sites used for deconvolution, including selection criteria and number; (c) assessment of confidence and assignment of tissue origins. Please also describe Issues, if any, with determining SNPs after bisulfite conversion.

We thank the reviewers for the suggestions.

a) In the revised manuscript, we have added Supplementary file 1 to list the sequencing depths of all the samples.

b) The criteria for selecting CpG sites for GETMap analysis are:

i) In the reference tissues, the difference between the highest and lowest methylation levels is greater than 25%; and

ii) After removing the tissue with either the highest or the lowest methylation level, the coefficient of variation (CV) of methylation levels across the remaining tissues is less than 0.3.

This information is added to the revised manuscript under the Materials and methods section.

We have also added the following information to Supplementary file 1:

i) The number of informative SNPs,

ii) The number of informative sequencing fragments,

iii) The number of informative CpG sites, and

iv) The number of CpG sites used for deconvolution.

c) To evaluate the accuracy of our approach, we performed the simulation analyses using GETMap to deconvolute 5 types of reference tissues including neutrophils, lymphocytes, lung, liver and placenta. Three sets of simulation analyses were performed to simulate the three clinical application scenarios in our study, namely pregnancy, transplantation and cancer. For each scenario, the numbers of informative DNA fragments, CpG sites and sequencing depth were matched with the median of the studied samples. Thirty independent simulations were performed for each scenario. The accuracy was calculated as the percentage contribution assigned to the tissue used for the deconvolution. For example, when the bisulfite sequencing data of liver tissue is used for deconvolution, the accuracy refers to the estimated contribution from the liver.

This information has been added to the revised manuscript (Results and Table 1).

In the study, GETMap analysis was performed on the bisulfite sequencing data of plasma DNA. As bisulfite treatment would convert unmethylated cytosines to thymine, differentiation between DNA fragments carrying C and T alleles would be less efficient. The differentiation of DNA fragments carrying C or T alleles on one strand would rely on the analysis of the complementary strand which carry G or A alleles, respectively. This discussion has been added to the revised manuscript.

2) The reference data for tissue-specific methylomes should be better described, including a demonstration of the specificity of tissue mapping.

In the revised manuscript, we have provided more detail information on the reference data of the tissue-specific methylomes as below:

“Whole-genome bisulfite sequencing data of 5 different tissues, including neutrophils, lymphocytes (combining B and T lymphocytes), liver, and lung were obtained from the BLUEPRINT Project (Martens and Stunnenberg, 2013), Roadmap Epigenomics (Roadmap Epigenomics Consortium et al., 2015), ENCODE (Davis et al., 2018) and GEO (Barrett et al., 2012). In addition, bisulfite sequencing data of two placenta tissues generated by our group were used as tissue-specific methylomes.”

3) The authors generally obtained genotyping data from the whole-genome arrays lacking low-frequency and rare genotypes. The authors should consider improving the performance of the method by increasing the number of informative genotypes using genotype imputation or whole-exome or whole-genome sequencing. This limitation could be addressed in the Discussion.

We thank the reviewers for the comments. In our study, both whole-genome arrays and whole genome sequencing were used to determine the genotypes of the study individuals. For the pregnancy and transplantation models, we used the Illumina microarray to identify the genotypes of the mother and fetus, as well as the lung transplant donors and recipients. This microarray platform targets both common and rare SNPs from the 1000 Genomes Project (minor allele frequency >2.5%). We obtained a median of 194,339 informative SNPs for the individuals in these two models. As shown in our simulation analysis, this number of informative SNPs would be enough for the downstream deconvolution analysis for samples in these two models. For the HCC study, we used whole-genome sequencing to identify cancer-associated mutations that were absent in the blood cells as the cancer-associated mutations would not be covered by the whole-genome arrays. This discussion has been included in the revised manuscript.